# An Economic Evaluation of the Adjuvanted Quadrivalent Influenza Vaccine Compared with Standard-Dose Quadrivalent Influenza Vaccine in the Spanish Older Adult Population

**DOI:** 10.3390/vaccines10081360

**Published:** 2022-08-20

**Authors:** Anna Fochesato, Sara Sottile, Andrea Pugliese, Sergio Márquez-Peláez, Hector Toro-Diaz, Ray Gani, Piedad Alvarez, Jesús Ruiz-Aragón

**Affiliations:** 1Department of Mathematics, University of Trento, Via Sommarive 14, 38123 Trento, Italy; 2Fondazione The Microsoft Research—University of Trento, Centre for Computational and Systems Biology (COSBI), Piazza Manifattura 1, 38068 Rovereto, Italy; 3Department of Economics, Faculty of Business, Universidad Pablo de Olavide, 41013 Sevilla, Spain; 4Evidence, Modeling and Synthesis, Evidera, Waltham, MA 02451, USA; 5Evidence, Modeling, and Synthesis, Evidera, London W6 8BJ, UK; 6Hospital de La Línea, 11300 La Línea de la Concepción, Spain

**Keywords:** influenza, vaccination, Spain, cost-effectiveness, adjuvanted, quadrivalent vaccine

## Abstract

Standard-dose quadrivalent influenza vaccines (QIV) are designed to provide protection against all four influenza strains. Adjuvanted QIV (aQIV), indicated for individuals aged 65+ years, combines MF59^®^ adjuvant (an oil-in-water emulsion of squalene oil) with a standard dose of antigen, and is designed to produce stronger and longer immune response, especially in the elderly where immunosenescence reduces vaccine effectiveness. This study evaluated the cost-effectiveness of aQIV vs. egg-based standard-dose QIV (QIVe) in the elderly population, from the payer and societal perspective in Spain. A dynamic transmission model, which accounts for herd protection, was used to predict the number of medically attended infections in Spain. A decision tree structure was used to forecast influenza-related costs and benefits. Influenza-related probabilities of outpatient visit, hospitalization, work absenteeism, mortality, and associated utilities and costs were extracted from Spanish and European published literature. Relative vaccine effectiveness (rVE) was sourced from two different meta-analyses: the first meta-analysis was informed by laboratory-confirmed influenza studies only, resulting in a rVE = 34.6% (CI95% 2–66%) in favor of aQIV; the second meta-analysis included real world evidence influenza-related medical encounters outcomes, resulting in a rVE = 13.9% (CI95% 4.2–23.5%) in benefit of aQIV. All costs were expressed in 2021 euros. Results indicate that replacing QIVe with aQIV in the Spanish elderly population would prevent on average 43,664 influenza complicated cases, 1111 hospitalizations, and 569 deaths (with a rVE = 34.6%) or 19,104 influenza complicated cases, 486 hospitalizations, and 252 deaths (with a rVE = 13.9%). When the rVE of aQIV vs. QIVe is 34.6%, the incremental cost per quality adjusted life years (QALY) gained was €2240 from the payer; from the societal perspective, aQIV was cost saving compared with QIVe. If the rVE was 13.9%, the incremental cost per QALY was €6694 and €3936 from the payer and societal perspective, respectively. Sensitivity analyses validated the robustness of these findings. Results indicate that replacing QIVe with aQIV in the Spanish elderly population is a cost-effective strategy for the Spanish healthcare system.

## 1. Introduction

Over the last few decades, vaccination has been a successful public health strategy to prevent various infectious diseases worldwide, effectively helping reduce the burden of vaccine-preventable diseases [1,2]. Influenza is an acute viral infection, highly transmissible, observed around the globe every year, with peak spread during the winter season [3]. Seasonal influenza is a vaccine-preventable disease. Influenza’s public health burden can be serious because of high transmissibility, accompanying comorbidities (e.g., pneumonia), and higher mortality, especially among the higher-risk population, such as the elderly [4]. Furthermore, in addition to disease management costs, influenza can have an increased socioeconomic impact due to productivity loses associated with missing work or absenteeism [5].

The clinical efficacy of vaccines against influenza has been improving steadily over the years, e.g., by going from trivalent to quadrivalent vaccines (the latter offers protection against all four viral strains), through the addition of vaccine adjuvants or by increasing antigen concentration [6]. Alongside the increased clinical efficacy, there is a growing amount of evidence on the cost-effectiveness of vaccination strategies against influenza, particularly if improved vaccines are used, such as quadrivalent influenza vaccines (QIV) [4].

Results from a recent modeling exercise across European settings support using enhanced influenza vaccines for the high-risk population (e.g., elderly patients) [7]. Economic modeling studies in Spanish settings have found cell-based QIVs (QIVc) to be cost-effective compared with traditional egg-based QIVs (QIVe) for adult patients (aged 9–64 years and at high-risk of complications) [8]; furthermore, for individuals aged 65 or older, adjuvanted QIV (aQIV) was cost-saving compared with high-dose (HD)-QIV [3], and adjuvanted trivalent influenza vaccine (aTIV) was found to be cost-effective compared with TIV [9,10].

Immunosenescence refers to the biological aging process associated with progressive decline in systemic immunity and increased prevalence of autoimmune and chronic diseases, increased vulnerability to common infectious, and poor responses to vaccination [11]. aQIV is indicated for individuals aged 65 years or older. aQIV combines MF59^®^ adjuvant (an oil-in-water emulsion of squalene oil) and a standard dose of antigen, and is designed to produce stronger and longer immune response, especially in the elderly where immunosenescence reduces vaccine effectiveness; compared with younger adults (18–64 years), vaccine effectiveness for the elderly was found to be 27% lower (37% for the elderly versus 51% for younger adults) [12]. Real-world evidence has shown that adjuvanted influenza vaccines results in statistically significantly fewer influenza-related medical encounters compared with non-adjuvanted influenza vaccines [12,13,14].

This study used a dynamic transmission model aimed to evaluate the cost-effectiveness of aQIV vs. QIVe in the elderly population (65+ years) in Spain. Given the dynamic nature of the model (i.e., accounting for indirect effect of vaccination), and similarly to several other models [15,16], the whole Spanish population is included so that herd protection can be accounted for; it has been reported that the indirect effects of vaccination can be more significant than the direct effects [17]. The model is used to project both costs and clinical benefits of competing vaccination strategies for the elderly, from the payer and societal perspectives.

## 2. Materials and Methods

### 2.1. Model Structure

The World Health Organization (WHO) guidelines were followed to conduct a cost-effectiveness analysis of the influenza vaccines in the Spanish older adult population [18] (The Drummond’s check-list for assessing economic evaluations is included as Appendix A). Influenza transmission and burden was simulated by adapting a Susceptible-Exposed-Infectious-Recovered (SEIR) model that was developed previously to evaluate the cost-effectiveness of aQIV in the Italian setting, and that was recently published [19,20]. The model was structured in two modules: epidemiological and disease burden.

The epidemiological module is a dynamic compartment model that allows to estimate the number of influenza cases by season. The output of the epidemiological model is the number of infections due to the influenza viral subtypes A(H1N1), A(H3N2), and B; both symptomatic and asymptomatic cases are predicted. The dynamic nature of the epidemiological module allows to incorporate the indirect effect of influenza vaccination, i.e., herd protection.

The burden module is a decision tree starting from the final output of the epidemiological model and simulating the natural history of the disease, i.e., among patients infected with influenza, the model estimates potential complications, which may require treatment, including hospitalization, and may also cause the subject’s death (Figure 1). The burden module model estimates the expected number of clinical events associated with influenza and the corresponding costs and quality-adjusted life years (QALY) estimated for each of the two influenza vaccines analyzed in the current assessment.

### 2.2. Epidemiological Module Inputs

#### 2.2.1. Population

The model was stratified into 86 age groups/classes. Contacts among age classes followed a published contact matrix for Spain, part of a larger study that analyzed 26 European countries [21]. The latent period was set to 1.5 days and the infectious period to 1.2 days; hence, the influenza generation time was 2.7 days [22].

The distribution of contact rates was chosen on the basis of the dominant (or codominant) strains in Spain in the years 2010–2019 [22,23]. In the absence of transmission rates for Spain, Italian transmission rates for the overall population were used as a proxy, using data for the same years in which matching strains were observed in the two markets (Table 1). In the only case in which the strains did not match (2017/2018 season), H3N2 data of the following year were used.

No estimates of R0 for influenza in Spain were identified, hence Italian estimates were used [22], assuming that in each season the contact rate for the dominant strains in Spain were the same as that of the corresponding strains in Italy. It should be noted that using the Spanish contact matrix and population structure combined with the contact rates estimated for Italy resulted in lower values of R0, and hence the average number of influenza infections predicted by the model per season would have been lower than published data for Spain [23]. Hence, a rescaled distribution of the Italian transmission rates was applied, using a factor of 1.05. This factor was found empirically to result in an average number of reported infections matching observed outcomes for Spain, reported in a study on transmissibility of influenza [23]. Furthermore, the 2014/2015 season (H3N2) was excluded from the analysis because the procedure estimating R0 for this season yielded a value below 1, in which case an epidemic cannot occur. A summary of the population included in the model is presented in Table 2. The Spanish population was sourced from the National Statistics Institute, Spanish Statistical Office (population reflects official data up to 2021) [24].

#### 2.2.2. Vaccine Coverage

Two different sources were used for vaccine coverage: one for individuals with ages between 0 and 64 (2012) [6], and another for those older than 65 years (2020) [25]. Coverage data for the 0–64 age category is aligned with a previous publication that evaluated the cost-effectiveness of quadrivalent influenza vaccine in Spain; the coverage for healthy individuals was reported as 0% (since in Spain vaccination is not recommended for healthy individuals) [6]. For the 65+ population, the model considers all individuals are at high risk for influenza infection, which is aligned with Spanish government guidelines for vaccination [26]. A summary of the coverage data used by the model is presented Table 3.

#### 2.2.3. Effectiveness

Estimates of QIVe effectiveness came from a recent systematic review and meta-analyses, which took into account differences between age groups and viral type. The meta-analysis results were summarized in a recent health-technology assessment (HTA) and are presented in Table 4 [20,27].

The relative vaccine effectiveness (rVE) of aQIV compared with QIVe was sourced from two published meta-analyses (Table 5). These meta-analyses reported on the rVE of aTIV vs. TIV. These results are extrapolated for the comparison of aQIV vs. QIV; the extrapolation is needed because to date, there are no real-world evidence aQIV studies [28]. Although the meta-analyses compared aTIV vs. TIV, these results are appropriate for aQIV as the European Medicines Agency (EMEA) established that observational effectiveness studies performed with aTIV are relevant to aQIV because both vaccines are manufactured using the same process and have overlapping compositions [29].

### 2.3. Disease Burden Module Inputs

#### 2.3.1. Rates of Clinical Events

Table 6 shows the probability of patients with a symptomatic case of influenza seeking different types of medical support. Visits to the general practitioner (GP) are further stratified as ambulatory (patient visits a doctor’s office) or a home visit (doctor visits the patient at home). Table 7 details the probability of developing complications and the distribution around the type of complications. These probabilities were not available from the literature specific to the Spanish settings, hence Italian data were used as a surrogate.

As patients experience influenza-related complications these may result in hospitalizations. Table 8 shows the probability of being hospitalized, and the distribution of hospitalizations by type of complication. Hospitalization data were not identified specific to the Spanish settings, hence Italian data were used as a proxy.

#### 2.3.2. Mortality

Only subjects who incur influenza-related complications face an influenza-specific risk of death. In absence of influenza-related death rates for Spain, data from the UK (previously used by Garcia et al. [6] when evaluating QIVs in Spain) and Italy were used. The risk of death (mortality likely attributable to influenza) was stratified by age categories and risk level (where applicable): ages 0–8 = 0.03%, 9–17 = 0.01%, 18–64 low risk = 0.15%, and high risk = 0.19%; 65+ = 2.67% [34,37].

#### 2.3.3. Costs

All costs are expressed in 2021 euros. It should be noted that in Spain, costs informed by the autonomous communities bulletins remain current from the time of their publication, until a new version is posted by the Spanish government, i.e., these official costs should not be inflated. The per-dose cost of each vaccine was €9.50 and €13 for QIVe and aQIV, respectively. These figures correspond to official tender prices set by the Spanish Ministry of Health [38]. The administration cost, evaluated from the public healthcare perspective, was €25.94 [39].

The disease management cost of influenza without complications considers the cost of GP visits (ambulatory or at home), priced at €59 and €83, respectively; the cost of pharmaceuticals at €3.21 (includes antivirals, drugs used for the symptomatic therapy of influenza, and antibiotics); and visits to the emergency room at €183 [9,40]. The disease management costs associated with ambulatory complications are detailed in Table 9. It should be noted that for individuals ≥18 years, there were data to inform the probability of different medical interventions in case of ambulatory complications following influenza; the probability of medical interventions was combined with unit local (Spanish) costs to inform the model. For the 0–17 age category however, detailed data on medical interventions were not available, hence overall costs were used.

Hospitalization costs by type of complication were as follows: upper-respiratory tract infection €2607.94; pneumonia €3393.23; chronic obstructive respiratory disease €3277.45; bronchitis €2507.91; and cardiac €3439.30 [41].

The model accounts for two categories of indirect costs: loss of productivity of workers due to influenza or resulting from premature death. Both categories of indirect costs have been estimated using the friction cost method [20]. The estimation of indirect cost combines the number of working hours per week (40) and the average pay per-hour (€17.34) [8], the employment rate (58.65% for 18–64 years individuals, 1.2% for those aged 65–69 years, and 0.3% for those aged 70+), sourced from official data from the Spanish Statistics National Institute [24]; the average number of working days lost for cases that did not require hospitalization (4.7 days) [42], and those that resulted in hospitalization (13.25) [35]; the probability of patients remaining at home as a result of developing influenza-like symptoms (48.32%) [42]; and probability of parents having to take care of sick children (35%) [8].

#### 2.3.4. Utilities

Table 10 shows the reference utilities (for healthy individuals) stratified by age categories [43,44]. As patients experience the disease, disutilities are applied associated with different clinical events. The disutility for influenza-related symptoms not requiring a medical visit was 0.005 [45]; for influenza-related symptoms requiring a GP visit was 0.06 [45]; for influenza-related symptoms with associated complications was 0.0075 [45]; and for cases requiring hospitalization, the disutility was 0.0090 [45]. Influenza-related disutilities were not available specifically for Spain, hence these were sourced from a burden of illness study sampling more than 2200 individuals from the general population in Belgium who had experienced influenza-like-illness [45].

As patients’ QALYs are accrued over several years in the future, the QALYs accrued after the first year were discounted at a 3.0% annual rate, following Spanish guidelines for health economics [46]. The same discount rate was used to accrue future indirect costs associated with averted deaths.

### 2.4. Analysis

The model allows for the calculation of burden of illness (i.e., number of symptomatic cases, medical help-seeking events with and without complications and with and without hospitalization, quality-adjusted life-years, and deaths), costs (direct: vaccine acquisition and administration and disease management, and indirect: productivity lost), and incremental analysis (cost per-QALY gained). From the public payer perspective, only direct medical costs are included, whereas for the societal perspective, the productivity lost costs were added to the direct medical costs. Incremental cost-effectiveness ratios (ICERs) were calculated to compare aQIV versus QIVe. The base case considers two main scenarios around relative vaccine effectiveness (rVE) of aQIV versus QIVe, given that two relevant published meta-analyses were identified providing quite different estimates for rVE: 34.6% and 13.9%.

Additional scenario analyses were conducted testing the impact of changes to model inputs and/or assumptions on the model results. A one-way deterministic sensitivity analysis (DSA) was used to identify the parameters that are key drivers of the ICER. In the DSA, parameters were changed using a ±20% variation from their base case value. A probabilistic sensitivity analysis (PSA) was also conducted by varying parameters based on their underlying probability distribution (the contact rate (beta) varied according to the expected distribution in the past 10 seasons; in alignment with previous publications, costs and disutilities were assumed to follow a gamma distribution using a coefficient of variation of 22%) [19,20]; 10,000 PSA iterations of the model were run to assess the effect of uncertainty on the ICERs (additional details regarding the sensitivity analysis are included in the Appendix B—Table A4, Table A5, Table A6, Table A7, Table A8, Table A9, Table A10 and Table A11). A willingness-to-pay threshold of €25,000 per QALY gained, relevant to Spain as per published literature, was used as the threshold to determine the cost-effectiveness of the interventions [47,48].

## 3. Results

The total number of people vaccinated was 9,150,385. Replacing QIVe with aQIV in the Spanish elderly population would on average prevent 43,664 influenza complicated cases, 1111 hospitalizations, and 569 deaths (with a rVE = 34.6%) or 19,104 influenza complicated cases, 486 hospitalizations, and 252 deaths (with a rVE = 13.9%). Additional details on the outcomes for the clinical events simulated are presented in Table 11. The incremental results for total costs and QALYs are shown in Table 12. Based on the incremental results, the ICER for the scenario using an rVE = 34.6% was €2240 per QALY gained from the payer perspective, and from the societal perspective aQIV was cost-saving; when using an rVE = 13.9%, the ICERs were €6694 and €3936 from the payer and societal perspectives, respectively. Therefore, using either estimate of relative-efficacy and from both perspectives, the results indicate that using aQIV as the vaccination strategy for the elderly population in Spain is cost-effective compared to QIVe. Table 13 shows the costs that the payer will incur for the vaccination of eligible individuals in Spain according to the population structure, vaccine coverage, and vaccines’ prices. Direct and indirect costs are detailed in the Appendix B (Table A1 and Table A2, respectively). Finally, QALYs stratified by age and type of clinical event are also presented in the Appendix B (Table A3).

Results from the DSA are presented in Figure 2 as tornado charts displaying the impact on the ICER of those parameters whose change caused the largest variations from the base-case ICER (from the payer’s perspective). DSA results are presented for both aQIV scenarios (i.e., for both estimations of rVE). Vaccine effectiveness, vaccine cost, and coverage were the most influential parameters in the dynamic model. Since aQIV is more effective than QIVe, a decrease of aQIV’s price makes it even more economically attractive than QIVe. In contrast, an increase in QIVe’s price impacts the cost-effectiveness analysis in the opposite way: since QIVe is less effective, increasing its price makes it less economically justifiable. It should also be noted that even for the scenarios with the largest variations in the ICERs, the value remained under the Spanish WTP threshold (€25,000 per QALY gained).

Results from the PSA indicate that from the payer’s perspective, the probability of aQIV being cost-effective compared with QIVe is 65% for the scenario using rVE = 34.6%, and 52.4% for the scenario using rVE = 13.9%. Figure 3 shows the results from the PSA as scatter plots (incremental QALYs versus incremental costs) from the payer perspective. Because of the nonlinearities of the system the majority of the simulations have a number of infected individuals lower than the mean.

## 4. Discussion

Given that healthcare payers have limited resources to fund the reimbursement of new healthcare interventions, including vaccination strategies against preventable infectious diseases, cost-effectiveness analyses are needed to support decision making regarding the use of these limited funding resources. Improvements in the effectiveness from new healthcare interventions typically come accompanied with a price premium to acquire these new interventions. From a payer’s perspective, it becomes critical to assess whether the higher cost of a new technology is worth paying when compared with its benefits. In most markets, new healthcare technologies undergo economic evaluations as one of several regulatory steps to achieve reimbursement, and to support pricing and purchasing decisions [49]. In particular, cost-effectiveness analyses are used regularly to assess the value of new vaccines and vaccination strategies [4].

The WHO provides recommendations twice a year regarding the composition of the vaccines for the influenza season, i.e., which virus strains should the influenza vaccines protect against [50]. Different vaccine types can then be selected matching the protection against particular viral strains recommended by the WHO. These different vaccines will have a range of effectiveness and be available at different prices, hence raising the question of which vaccines should be preferred by a payer interested in getting the best possible value from their investment, potentially stratifying the decision by subpopulations, e.g., by age, or depending on the level of risk experienced. Even within the same class of vaccines (e.g., among quadrivalent vaccines), there will be different options to select from. This analysis focused on replacing QIVe with aQIV in the Spanish elderly population, making use of a dynamic transmission model that allowed accounting for indirect (herd) protection across the entire Spanish population.

A majority of published economic analysis reports adult vaccinations strategies to be cost-effective [51]. In alignment with prior literature, results from the present study indicate that replacing QIVe with aQIV in the Spanish elderly population is a cost-effective strategy. The cost-effectiveness result holds even when using a lower rVE (i.e., 13.9% vs. 34.6%) for aQIV. The lower rVE was informed by published relative effectiveness outcomes that included different real world evidence influenza-related medical encounters outcomes, as complementary to laboratory-confirmed influenza studies only. Although the acquisition cost of aQIV is higher compared with QIVe (27% higher), the better effectiveness of aQIV (using estimates from both meta-analyses) results in cost saving on disease management and better clinical outcomes which translate into increased QALYs. The net effect of increased total costs accompanied by QALY gains results in ICERs well below the cost-effectiveness threshold deemed to be appropriate in Spain (€25,000 per QALY gained). The elderly population have a higher risk of experiencing the worse clinical outcomes derived from influenza, hence offering elderly patients the most effective vaccine results in tangible clinical and economic benefits.

A key strength of the analysis was the use of a dynamic model, making it possible to account for herd protection within the elderly group and other age groups of the Spanish population (<65yrs). Vaccination against infectious diseases has an indirect benefit on non-vaccinated individuals, a benefit that cannot be captured by static models. In fact, using dynamic models is currently recommended as the framework to be used for economic evaluations of vaccines [18].

Conducting the analysis faced several limitations as well. First, although every attempt was made to inform the model with data that was specific to the Spanish settings, this was not always possible due to the paucity of data. This was the case for disease transmission rates, where Italian data were used as a proxy; the viral strains circulating in both countries were similar most seasons, however, which should increase the validity of the approach. Italian data were also used to inform R0, and upon further examination, a rescaled distribution of the Italian transmission rates was used to better match the number of infections predicted by the model with published figures for Spain. Italian data were also used to inform the rates of clinical events among patients with symptomatic influenza. Second, a key model parameter, rVE, was subject to relatively high uncertainty as two published meta-analyses proposed rather different estimates for rVE. Furthermore, the meta-analyses used provided a comparison of aTIV vs. TIV that was extrapolated to be used for aQIV; given that these two vaccines (aTIV and aQIV) have overlapping compositions and undergo similar manufacturing processes, the extrapolation has been deemed appropriate by the EMEA. Nonetheless, analysis using either estimation of rVE resulted in aQIV being cost-effective (or cost saving) from both perspectives (payer and societal), hence even using the more conservative estimation of relative effectiveness, the conclusion is still that it is worth replacing QIVe with aQIV for the elderly population in Spain. In spite of these limitations, the model clinical results were validated versus relevant published results for Spain, hence increasing the model’s face validity. In fact, results from the current analysis are aligned with prior analysis mentioned earlier that were conducted for Spain in which aTIV was found to be cost-effective compared with TIV [9,10].

## 5. Conclusions

Considering the potential negative impacts on clinical outcomes and disease management that influenza can have in the elderly population, and the benefits derived from the use of an influenza vaccine with better relative effectiveness, results for the analysis have shown that aQIV represents an affordable and highly cost-effective alternative to vaccinate the elderly in Spain. Results from these analyses should help inform regional decision makers in Spain as they determine which vaccination strategies should be funded that will provide the highest health outcomes for the older adult population.

## Figures and Tables

**Figure 1 vaccines-10-01360-f001:**
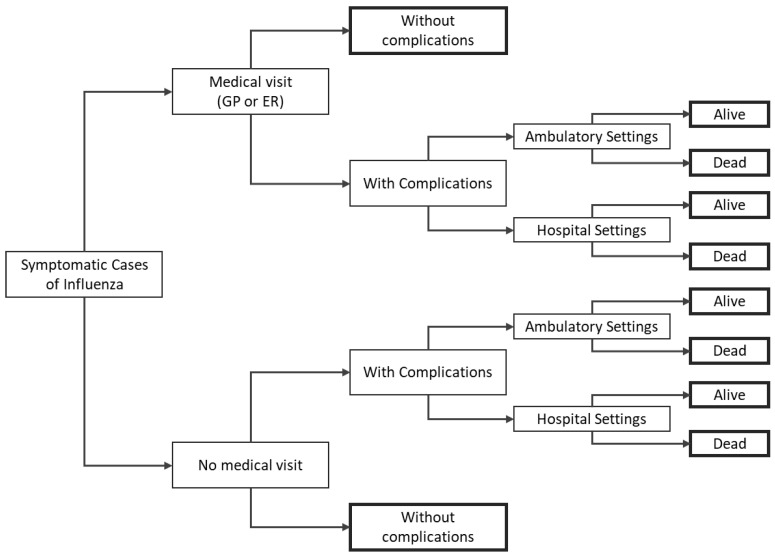
Decision-tree structure for the disease burden module. Note: bold rectangles represent terminal nodes. ER = emergency room; GP = general practitioner.

**Figure 2 vaccines-10-01360-f002:**
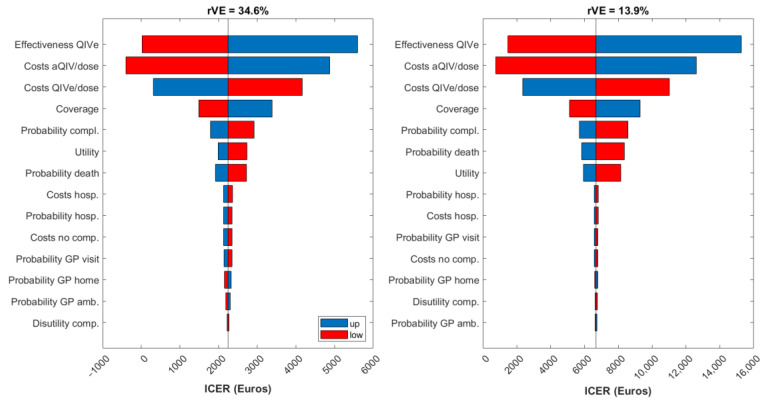
Tornado diagrams for the ICER from the payer perspective. aQIV = adjuvanted quadrivalent influenza vaccine; amb. = ambulatory; comp. = complications; hosp. = hospitalization; QIVe = standard-dose quadrivalent influenza vaccine; rVE = relative vaccine effectiveness.

**Figure 3 vaccines-10-01360-f003:**
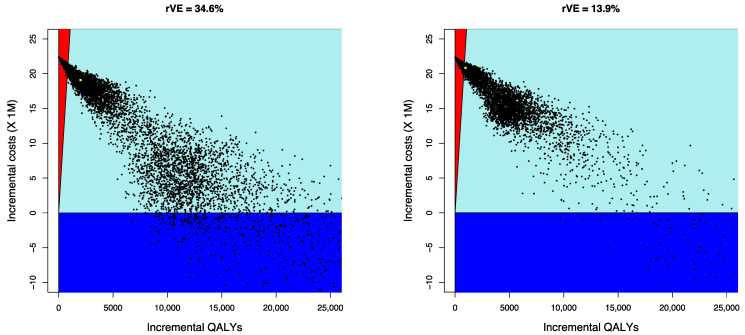
Cost-effectiveness scatter plots—incremental QALYs versus incremental costs (payer perspective). Note: dark blue represents cost-saving scenarios, the light blue sector represents results under €25,000/QALY; red represents results over €25,000/QALY; aQIV = adjuvanted quadrivalent influenza vaccine; QALY = quality-adjusted life-years.

**Table 1 vaccines-10-01360-t001:** Influenza strains for influenza seasons after 2010/11.

*Season*	*Strain Circulation in Spain*	*Strain Circulation in Italy*
2010/11	**H1N1**	**H1N1**/B
2011/12	**H3N2**	**H3N2**
2012/13	**B**	H1N1/**B**
2013/14	**H1N1/H3N2**	**H1N1/H3N2**
2014/15	**H3N2**	H1N1/**H3N2**/B
2015/16	**H1N1**	**H1N1**/H3N2/B
2016/17	**H3N2**	**H3N2**
2017/18	**B**/H3N2	H1N1/**B**
2018/19	Not available	H1N1/H3N2

Note: Bold font highlights the correspondence of strains between Spain and Italy by season. Underlined text indicates data from a different season used as a proxy.

**Table 2 vaccines-10-01360-t002:** Summary of Spanish population structure.

*Age*	*N*	*% Healthy*	*% at Risk*	*At-Risk—Influenza Coverage*	*Overall—Influenza Coverage*
0–8	3,662,079	79.93%	20.07%	24.20%	4.86%
9–17	4,500,901	78.63%	21.37%	24.24%	5.18%
18–64	29,719,673	57.46%	42.54%	17.15%	7.29%
≥65	9,444,037	0	100%	67.7%	67.7%

Note: Based on the Spanish Ministry of Health, the ‘at-risk’ population included individuals with chronic cardiovascular or lung disease, metabolic disease, morbid obesity, chronic renal disease, hemoglobin disorders and anemia, asplenia, chronic liver disease, severe neuromuscular diseases, immunosuppressed, cochlear implanted, cognitive dysfunction, people living in closed institutions, pregnant women, and children from 6 months to 18 years receiving long-term treatment with acetylsalicylic acid. Individuals without these conditions are considered healthy, i.e., influenza vaccination is not recommended to them [6].

**Table 3 vaccines-10-01360-t003:** Overall vaccine coverage.

*Age Category*	*0–4*	*5–17*	*18–49*	*50–64*	*65–69*	*70–74*	*75–79*	*80–84*	*85+*
Influenza vaccine coverage	4.55%	5.18%	2.91%	15.66%	59.84%	67.41%	68.36%	76.39%	72.23%

**Table 4 vaccines-10-01360-t004:** QIVe absolute vaccine effectiveness.

*Age Category*	*Viral Strain*		
	*H1N1*	*H3N2*	*B*
0.5–1	69.0 (49.0–81.0)	43.0 (28.0–55.0)	66.5 (57.7–73.6)
2–6	69.0 (49.0–81.0)	43.0 (28.0–55.0)	66.5 (57.7–73.6)
7–17	73.0 (52.0–84.0)	35.0 (14.0–41.0)	77.0 (18.0–94.0)
18–64	73.0 (49.0–81.0)	35.0 (14.0–41.0)	77.0 (18.0–94.0)
≥65	62.0 (36.0–78.0)	24.0 (−6.0–45.0)	52.1 (41.5–60.8)

Note: Effectiveness reported as mean% (95% confidence interval).

**Table 5 vaccines-10-01360-t005:** aQIV relative vaccine effectiveness (extrapolated from aTIV).

*Meta-Analysis*	*rVE*	*Notes*
Calabrò et al., 2021 [20]	34.6% (95% CI: 2.0–66.0%)	Synthetized three studies that reported the relative effectiveness of aTIV against TIV, based on laboratory-confirmed influenza studies.
Coleman et al., 2021 [12]	13.9% (95% CI 4.2–23.5%)	Studied the effectiveness of aTIV relative to vaccination with TIV. It included influenza-like-illness outcomes using influenza-related medical encounters for influenza with or without pneumonia in various clinical settings including outpatient, hospital, or emergency department.

aQIV = adjuvanted quadrivalent influenza vaccine; aTIV = adjuvanted trivalent influenza vaccine; CI = confidence interval; rVE = relative vaccine effectiveness; TIV = trivalent influenza vaccine.

**Table 6 vaccines-10-01360-t006:** Medical support seeking by patients with a symptomatic case of influenza.

*Age Category*	*Probability of Medical Support Seeking, by Type*			
	*GP Visit* [30]	*GP Ambulatory* [31]	*GP Home Visit* [31]	*Emergency Room* [31,32]
0–8	65.63%	34.02%	65.98%	3.04%
9–17	57.63%	34.02%	65.98%	1.65%
18–64	32.03%	34.02%	65.98%	0.02%
≥65	36.89%	34.02%	65.98%	0.02%

GP = general practitioner. Note: ambulatory and home visit correspond to the distribution of GP visit types; i.e., 34% of the total GP visits are considered ambulatory, whereas ~66% are home visits.

**Table 7 vaccines-10-01360-t007:** Influenza-related complications.

*Age Category*	*Probability of Influenza-Related Complications* [31,33,34]	*Distribution of Influenza Complications* [31,32]			
		*URTI*	*Bronchitis*	*Pneumonia*	*Other Respiratory*
**0–8**	22.21%	54.46%	43.31%	2.23%	0.00%
9–17	15.09%	54.55%	43.64%	1.82%	0.00%
18–64 LR	29.98%	52.33%	39.52%	3.63%	4.52%
18–64 HR	55.33%	52.33%	39.52%	3.63%	4.52%
≥65	63.65%	52.33%	39.52%	3.63%	4.52%

HR = high risk; LR = low risk; URTI = upper respiratory tract infection.

**Table 8 vaccines-10-01360-t008:** Influenza-related hospitalizations.

		*Hospitalizations by Complications* [35,36]					
*Age Category*	*Probability Hosp.* [31,32]	*URTI*	*Bronchitis*	*Pneumonia w/o Comp.*	*Pneumonia with Comp.*	*COPD*	*Cardiac*
0–8	4.14%	0.00%	0.00%	100.00%	0.00%	0.00%	0.00%
9–17	2.73%	0.00%	0.00%	100.00%	0.00%	0.00%	0.00%
18–64 LR	0.41%	23.53%	5.88%	29.41%	41.18%	0.00%	0.00%
18–64 HR	2.96%	23.53%	5.88%	29.41%	41.18%	0.00%	0.00%
≥65	2.96%	15.38%	3.85%	19.23%	26.92%	19.23%	15.38%

COPD = chronic obstructive respiratory diseases; comp. = complications; hosp. = hospitalization; HR = high risk; LR = low risk; URTI = upper respiratory tract infection; w/o = without.

**Table 9 vaccines-10-01360-t009:** Ambulatory complications cost.

*Age Category*	*Resource*	*Probability (%)*	*Cost*
0–17	URTI costs	N/A	€59.00 [40]
	LRTI costs		€171.45 [39]
≥ 18 [40]	Antibiotic treatment (×5 days)	95.48%	€3.00
	Specialist visit	1.04%	€215.00
	X-ray thorax	7.72%	€23.34
	X-ray sinuses	0.52%	€23.34
	X-ray others	0.28%	€23.34
	Hematology	0.61%	€4.00
	ECG	0.24%	€15.00
	Blood analysis	0.09%	€5.00
	Throat swab	0.05%	€18.00
	Audiometry	0.05%	€62.00

ECG = electrocardiogram; LRTI = lower respiratory tract infections; N/A = not applicable; URTI = upper respiratory tract infections.

**Table 10 vaccines-10-01360-t010:** Utilities (for healthy individuals).

*Age Category*	*Utility* [43,44]
0–8	0.95
9–17	0.95
18–64	0.93
≥65	0.87

**Table 11 vaccines-10-01360-t011:** Clinical events prevented—aQIV vs. QIVe.

*Age Category*	*Medical Visits without Complications*		*Medical Visits with Complications*		*Hospitalizations*		*Deaths*	
	*aQIV rVE = 34.6%*	*aQIV rVE = 13.9%*	*aQIV rVE = 34.6%*	*aQIV rVE = 13.9%*	*aQIV rVE = 34.6%*	*aQIV rVE = 13.9%*	*aQIV rVE = 34.6%*	*aQIV rVE = 13.9%*
0–8	7253	3110	3015	1293	124	53	1	1
9–17	6964	3021	2089	906	57	25	1	1
18–64	8833	3847	18,529	8070	338	147	32	14
≥65	4221	1862	20,031	8835	592	261	535	236
Total	27,271	11,840	43,664	19,104	1111	486	569	252

aQIV = adjuvanted quadrivalent influenza vaccine; QIVe = standard-dose quadrivalent influenza vaccine.

**Table 12 vaccines-10-01360-t012:** Incremental costs and QALYs results aQIV vs. QIVe.

*Age Category*	*Direct Medical Costs (Thousands)*		*Indirect Costs (Thousands)*		*QALYs*	
	*rVE = 34.6%*	*rVE = 13.9%*	*rVE = 34.6%*	*rVE = 13.9%*	*rVE = 34.6%*	*rVE = 13.9%*
0–8	−€1497	−€642	−€1905	−€1023	112.8	64.9
9–17	−€1074	−€467	−€1294	−€828	109.8	63.4
18–64	−€2540	−€1109	−€13,507	−€5905	992.4	433.6
≥65	€19,224	€20,990	−€1103	−€487	5083.8	2242.6
Total	€14,112	€18,773	−€17,808	−€8243	6298.7	2804.5

aQIV = adjuvanted quadrivalent influenza vaccine; QIVe = standard-dose quadrivalent influenza vaccine; QALY = quality-adjusted life years; rVE = relative vaccine effectiveness (aQIV vs. QIVe).

**Table 13 vaccines-10-01360-t013:** Vaccine acquisition and administration costs.

*Age Category*	*Administration (Thousands)*	*Acquisition (Thousands)*	
		*QIVe*	*aQIV*
0–8	€9227	€3379	€3379
9–17	€6048	€2215	€2215
18–64	€56,236	€20,595	€20,595
≥65	€165,850	€60,739	€83,117
Total	€237,361	€86,929	€109,306

aQIV = adjuvanted quadrivalent influenza vaccine; QIVe = standard-dose quadrivalent influenza vaccine.

## Data Availability

All data used in the analysis is included in the publication.

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
