# Peer review of "An Economic Evaluation of the Adjuvanted Quadrivalent Influenza Vaccine Compared with Standard-Dose Quadrivalent Influenza Vaccine in the Spanish Older Adult Population"

_vaccines, 2022, doi:10.3390/vaccines10081360_

Round 1
Reviewer 1 Report
This is a very interesting paper insofar as it goes beyond the biological evaluation of vaccines considering not only the improved efficiency moving from the trivalent to quadrivalent vaccines against influence but also the relation between the improved efficiency and the increased cost. In other words, this manuscript is a cost-effectiveness analysis. The evaluation of the benefits is done considering also the special case of older adults who require the use of adjuvanted influenza vaccines, designed to overcome immune senescence.
I find very interesting the dynamic nature of their analysis and the observation that the dynamic nature of their modelling makes it possible for them to establish the indirect benefit on non-vaccinated individuals.
For these reasons I warmly recommend this paper for publication.
In Fig. 2 I noticed an inversion of color moving from Costs aQIV dose to Costs QIVe/dose. What is the meaning of this inversion? Perhaps the authors have an easy explanation that may improve this excellent paper.
Author Response
Point 1: In Fig. 2 I noticed an inversion of color moving from Costs aQIV dose to Costs QIVe/dose. What is the meaning of this inversion? Perhaps the authors have an easy explanation that may improve this excellent paper.
Response: As reported in the manuscript aQIV is cost-effective compared with QIVe. Hence, increasing the acquisition cost of QIVe further improves the ICER, whereas increasing the acquisition cost of aQIV has the opposite effect, worsening the ICER.
See text lines 313-317 in the updated manuscript for an enhanced explanation around this.
Reviewer 2 Report
In the current well executed study, the authors use mathematical modeling to predict the cost savings associated with improving quadrivalent inactivated influenza vaccination effectiveness with the co-administration of an adjuvant.
The study is timely as means to improve the efficacy of influenza vaccines are needed.
Concerns regarding the manuscript in its current form are minor and are the following:
1. The final paragraph of the introduction needs to be improved in organization to more effectively transition to the discussion of the elderly and clearly state the goals of the study.
2. The terms DSA, PSA, ICER require definition.
3. Figure 2, the tornado plot is difficult to read and does not add meaningful to the manuscript. The text is too small as presented and could be improved by having both effectiveness compared in one plot. Alternatively, the figure could be removed.
4. The manuscript contains an abundance of tables (16 in total) of information required to describe the model as well as the results. Some of these should be moved to an appendix so that the tables of key importance are not diluted.
5. Line 413 contains "vaccine" which should be "vaccinate".
Author Response
Point 1. The final paragraph of the introduction needs to be improved in organization to more effectively transition to the discussion of the elderly and clearly state the goals of the study.
Response 1. See text lines 76-81 in the updated manuscript for changes made to the introduction addressing this comment.
Point 2. The terms DSA, PSA, ICER require definition.
Response 2. These terms are defined when first used in the manuscript:
DSA: text line 272; PSA: text line 274; ICER: text line 265.
Point 3. Figure 2, the tornado plot is difficult to read and does not add meaningful to the manuscript. The text is too small as presented and could be improved by having both effectiveness compared in one plot . Alternatively, the figure could be removed.
Response 3. The figure has been replaced by a new one that uses a bigger font size to improve readability.
Note that both measures of relative vaccine effectiveness (rVE) for aQIV vs QIVe cannot be represented within a single tornado plot, since using each rVE generates different base case results.
Point 4. The manuscript contains an abundance of tables (16 in total) of information required to describe the model as well as the results. Some of these should be moved to an appendix so that the tables of key importance are not diluted.
Response 4. We recognize that the manuscript includes as many tables as described, intended to provide quick access to the inputs that were used in the analyses.
In the updated manuscript we have moved three tables (originally Table 14, 15 and 16) to the supplemental appendix.
Point 5. Line 413 contains "vaccine" which should be "vaccinate".
Response 5. This typo has been fixed (text line 424).
Reviewer 3 Report
The title of the paper and the abstract did not reflect the study presented. Moreover, the model presented is based on market vaccine data from Spanish older adult population but data analysis are referring all ages. Besides the vaccine proposed only is licensed for 65+ populations. Is not clear from data and model analysis that a replacement of vaccine with other will result in economical benefit, instead the vaccination will do it! Moreover, the data analysis did not compare to types of vaccines to be used for validation of author’s conclusions, neither vaccines results were presented for comparison.
I my opinion the paper is not with quality to be consider for publication.
Author Response
Point 1. The title of the paper and the abstract did not reflect the study presented. Moreover, the model presented is based on market vaccine data from Spanish older adult population but data analysis are referring all ages. Besides the vaccine proposed only is licensed for 65+ populations. Is not clear from data and model analysis that a replacement of vaccine with other will result in economical benefit, instead the vaccination will do it! Moreover, the data analysis did not compare to types of vaccines to be used for validation of author’s conclusions, neither vaccines results were presented for comparison.
Response. We are sorry that the analysis performed was not sufficiently clear. We have added text lines 76-81 to the introduction, pointing out early on the dynamic nature of the model being used, and hence the need to include the whole Spanish population.
Indeed our work is based on using a dynamic model of infectious disease transmission. A dynamic transmission model estimates the risk of infection as a function of the number of infected individuals in the population at a given point in time. If a vaccine reduces the number of infections, then the associated risk of infection across the whole population (not only those individuals who have been vaccinated) will decrease. Because dynamic models estimate direct and indirect effects that may arise from a vaccination program, the full population of interest should be included.
The two vaccination scenarios analyzed in the model differ from each other specifically in the use of different vaccines for the elderly (65+ population), while in the other age groups the vaccination scheme remains the same. However, the change in vaccination strategy in the 65+ population affects other age groups (due to the nature of contacts between individuals, captured by the dynamic model); this is why the model had to consider the entire population.
Furthermore, note in Table 13, reporting on the total vaccine acquisition costs, that the total costs under QIVe and aQIV are the same across age categories, except for the elderly category (65years). This is because it is only for the elderly population that a different vaccine is considered. In fact, from the total acquisition cost, vaccination with aQIV is more expensive. The additional acquisition cost of aQIV is offset by savings in direct medical costs/disease management, since using aQIV means that clinical events of interest are prevented (e.g., medical visits, hospitalizations) compared to using QIVe.
Round 2
Reviewer 3 Report
The authors' answers regarding the relevance of the analysis presented in this article, which I consider not very enlightening, so I maintain my opinion.
Author Response
Point 1: The authors' answers regarding the relevance of the analysis presented in this article, which I consider not very enlightening, so I maintain my opinion.
Response 1: Honestly, we do not quite understand which exactly are the objections of the Reviewer to our approach.
We have added a few citations to lines 79-80, in which it is shown how the approach we used is similar to many others in the literature, and the need is emphasized of taking into account indirect effects of vaccination.
We hope that this may clarify the soundness of our approach.